The association of plasma osteoprotegerin levels and functional outcomes post endovascular thrombectomy in acute ischemic stroke patients: a retrospective observational study

Park Moo-Seok 1
Park Jin-Hee 2
Joo Ahran 3
Chang Yoonkyung tin1207@nate.com 4
Song Tae-Jin knstar@hanmail.net 1
1 Department of Neurology, Seoul Hospital Ewha Womans University College of Medicine , Seoul , Republic of Korea
2 Department of Molecular Medicine, College of Medicine, Graduate Program in System Health Science and Engineering, Ewha Womans University , Seoul , Republic of Korea
3 Department of Medicine, Ewha Womans University College of Medicine , Seoul , Republic of Korea
4 Department of Neurology, Mokdong Hospital Ewha Womans University College of Medicine , Seoul , Republic of Korea
Musa Kamarul Imran
Electronic publication date: 2022 May 3
Publication date: 2022
Volume: 10
Electronic Location ID: e13327
Received 2021 Jun 9; Accepted 2022 Apr 4
Copyright: ©2022 Park et al.
Copyright year: 2022
Copyright holder: Park et al.
License: This is an open access article distributed under the terms of the Creative Commons Attribution License, which permits unrestricted use, distribution, reproduction and adaptation in any medium and for any purpose provided that it is properly attributed. For attribution, the original author(s), title, publication source (PeerJ) and either DOI or URL of the article must be cited.
License URL: https://creativecommons.org/licenses/by/4.0/

Keywords: Osteoprotegerin, Stroke, Prognosis, Endovascular thrombectomy

Funding: The Basic Science Research Program through the National Research Foundation of Korea funded by the Ministry of Education 2021R1F1A1048113 2021R1I1A1A01059868 This project was supported by the grant from the Basic Science Research Program through the National Research Foundation of Korea funded by the Ministry of Education (2021R1F1A1048113 to Tae-Jin Song, 2021R1I1A1A01059868 to Yoonkyung Chang). The funders had no role in study design, data collection and analysis, decision to publish, or preparation of the manuscript.

==============================
Background

Osteoprotegerin (OPG), also known as osteoclastogenesis inhibitory factor, is a tumor necrosis factor receptor superfamily component. There is an established relationship between OPG and cardiovascular disease. We hypothesized that plasma OPG levels are associated with functional outcomes in acute ischemic stroke patients who have undergone endovascular thrombectomy (EVT).

Methods

From April 2014 through December 2020, a total of 360 acute ischemic stroke patients who underwent EVT were prospectively included in this retrospective observational study. Plasma OPG was measured after fasting for 12 postoperative hours after EVT. A modified Rankin Scale (mRS) was used to assess functional outcomes 3 months after index stroke occurrence. Univariate and multivariate binary logistic regression and ordinal logistic regression analyses were performed to investigate the association of plasma OPG levels with poor functional outcomes.

Results

Overall, 145 (40.2%) patients had poor (mRS > 2) outcomes. The mean ± standard deviation plasma OPG level was 200.2 ± 74.4 pg/mL. Multivariate analysis after adjusting for sex, body mass index, and variables with p < 0.1 in the preceding univariate analysis revealed high plasma OPG levels were independently associated with poor functional outcomes (highest tertile vs. lowest tertile of OPG; odds ratios (OR) 2.121, 95% confidence interval (CI) [1.089–4.191], p = 0.037 in binary logistic regression, OR 2.102, 95% CI [1.301–3.412], p = 0.002 in ordinal logistic regression analysis).

Conclusions

This study demonstrated that higher plasma OPG levels were associated with poor functional outcomes in acute ischemic stroke patients who underwent EVT.

Introduction

In acute ischemic stroke management, endovascular thrombectomy (EVT) is widely practiced and supported by evidence from systematic reviews of randomized controlled trials (class 1A evidence). Recently, a time window of up to 24 h for EVT implementation has been suggested (Powers et al., 2015; Powers et al., 2018). It is important for clinicians or neurointerventionalists to predict prognosis when performing EVT on acute ischemic stroke patients. Early and successful recanalization of occluded blood vessels with the presence of good collaterals have been associated with functional outcomes among acute ischemic stroke patients after EVT (Goyal et al., 2019). However, more research is needed regarding biomarkers associated with the prognosis after EVT.

Osteoprotegerin (OPG), a component of the tumor necrosis factor (TNF) receptor superfamily, functions as a decoy receptor for receptor activator of nuclear factor- κB ligands, and it inhibits the apoptosis of specific cells by binding to TNF-related apoptosis ligands (Simonet et al., 1997; Yasuda et al., 1998). OPG is elicited from vascular endothelia or smooth muscle cells, and it regulates inflammatory processes and vascular injury (Zannettino et al., 2005). Moreover, these inflammatory reactions among stroke patients are closely associated with functional outcomes assessed at 3 months after stroke by the modified Rankin Scale (mRS) (Chen et al., 2021; Li et al., 2016; Oh et al., 2020).

Endothelial dysfunction or damage could occur after performing EVT (Gory et al., 2013; Rochette et al., 2019; Teng et al., 2015). This endothelial dysfunction or damage is related to the prognosis of patients after EVT (Seo et al., 2017; Stepien et al., 2012). Therefore, there is a possibility that the plasma OPG level representing endothelial dysfunction is positively correlated with functional outcomes in stroke patients who have undergone EVT. Previous studies have suggested an association between plasma OPG levels and cardiovascular disease. Elevated plasma levels of OPG are associated with coronary artery atherosclerosis (Rhee et al., 2005), carotid artery stenosis (Kadoglou et al., 2008), and peripheral artery atherosclerosis (Chen et al., 2017; Helske et al., 2007). In stroke patients, plasma OPG levels are associated with the burden of cerebral atherosclerosis and poor prognosis (Kim et al., 2013; Nezu et al., 2015; Song et al., 2012; Wu et al., 2016). However, little is known about the association between OPG levels and functional outcomes in acute ischemic stroke patients after EVT. After performing the EVT, endothelial dysfunction or damage could have occurred (Gory et al., 2013; Rochette et al., 2019; Teng et al., 2015). This endothelial dysfunction or damage is related to the prognosis of patients after EVT (Seo et al., 2017; Stepien et al., 2012). Therefore, we hypothesized that the plasma OPG level representing endothelial dysfunction or damage would be associated with functional outcomes in stroke patients who have undergone EVT.

Materials & Methods

Study design

This study was a retrospective observational study. Our institutional review board approved this study, and we received informed consent from all patients or their closest relatives (EUMC 2014-04-023, 2018-10-036). The informed consent during the data collection process was explained directly to the patient by the researcher and was then signed. Withdrawal of consent was possible at any time, and blood samples from the patients who withdrew consent were not included in the final analysis. After collecting the research data, data management and analysis were performed in an anonymized state by entering the patient’s de-identified study number.

Subjects

From April 2014 through December 2020, 413 patients diagnosed with acute ischemic stroke and who underwent EVT were prospectively screened from Ewha Womans University Seoul Hospital and Ewha Womans University Mokdong Hospital in Korea. All screened patients were (1) admitted within 24 h after the onset of neurological symptoms or the last known normal time, (2) diagnosed with acute ischemic stroke confirmed by brain computed tomography (CT) or magnetic resonance image (MRI), and (3) performed EVT after confirming vessel occlusion and perfusion-diffusion mismatch. We excluded patients who had uncommon stroke risk factors, including cancer (n = 8), autoimmune disease (n = 2), or a history of bone fracture (n = 4) within the last 2 months because OPG is closely involved with systemic inflammatory reactions and bone metabolism (Kiechl et al., 2006; Song et al., 2012). Any samples that could not be analyzed for this study due to poor blood sampling quality were excluded (n = 1), and patients who refused (n = 33) or withdrew consent were also excluded (n = 5) (Fig. 1). Finally, we included 360 patients in this study and collected blood samples for plasma OPG measurement after fasting for 12 postoperative hours after EVT.

Figure 1 Flow chart of the study design.

We evaluated all patients using our institution’s standard stroke evaluation protocol based on the American Heart Association and Korean clinical practice guidelines (Jauch et al., 2013; Ko et al., 2019), which included routine blood tests (blood glucose level at admission, total cholesterol, low-density lipoprotein, triglyceride, white blood cell count, hemoglobin, creatinine, total calcium, phosphate, albumin, C-reactive protein, and vitamin D 25(OH)D), chest radiography, 12-lead electrocardiography, and brain imaging including CT and CT angiography and/or MRI and MR angiography or digital subtraction angiography (DSA) (Chang et al., 2018).

Clinical and radiologic assessment

Detailed information regarding risk factors is shown in the supplementary methods (Chang et al., 2018). The National Institute of Health Stroke Scale (NIHSS) was used to evaluate the severity of neurological symptoms. Successful recanalization was defined as a grade of 2b or 3, assessed by modified treatment in cerebral ischemia based on the final DSA (Higashida et al., 2003). Hemorrhagic transformation was defined according to the European Cooperative Acute Stroke Study (ECASS)-3 trial classification and was confirmed in follow-up brain images by consensus discussions (Hacke et al., 2008; Higashida et al., 2003). Stroke subtypes were classified into three categories according to the Trial of Org 10172 in Acute Stroke Treatment classification system (Adams Jr et al., 1993).

Acute stroke treatment

Administration of intravenous tissue plasminogen activator (IV tPA) and EVT were performed according to current guidelines for acute stroke management (Hacke et al., 2008). IV tPA administration was performed under the supervision of a neurologist according to standard guidelines within 4.5 h of symptom onset (Hacke et al., 2008). EVT also adhered to standard guidelines, and the decision on whether to perform EVT was arrived at through discussions between neurologists and neuroradiologists (Jauch et al., 2013; Kernan et al., 2014). If the appropriate time window for performing EVT was uncertain, CT or MR angiographic, brain MR diffusion-perfusion, diffusion-fluid attenuated inversion recovery (FLAIR), and diffusion-clinical mismatch findings were considered to inform the decision (Jauch et al., 2013; Kernan et al., 2014). EVT primarily involved mechanical thrombectomy. If reocclusion or distal embolization was present, glycoprotein IIb/IIIa antagonists could be administered as adjuvant therapy. Noncontrast brain CT was routinely performed 24 h after EVT, and brain MR (diffusion, FLAIR, gradient recalled echo image) and MR angiography were also routinely performed after 48–72 h.

Outcomes

In the present study, a modified Rankin Scale (mRS) at 3 months (90 days ± 14 days) after index stroke occurrence was used to assess functional outcomes. The mRS was evaluated by a neurologist with more than 5 years of experience and assessed based on face-to-face visits when the patients visit the outpatient clinic; if they could not visit in person, we provided a teleconsultation. Poor outcomes were defined by mRS ratings > 2.

Measurement of plasma OPG levels

At admission, we collected venous samples in EDTA from all enrolled patients. Samples were immediately centrifuged at 1,900 g for 15 min at the correct temperature (4 °C), followed by storing at −80 °C until analysis. OPG DuoSet (R&D Systems, Abingdon, UK), an enzyme-linked immunosorbent assay kit, was used to measure plasma OPG levels. The detection range was 31.25–4000 pg/mL. One of our research team members measured and averaged OPG levels in duplicate (Song et al., 2012). Coefficients of variability were 3.1% for inter-assay comparisons and 2.5% for intra-assay comparisons.

Sample size estimation

According to previously published studies, the rates of poor functional outcome defined as below mRS 2 are reported to be between 20 and 40% (Appelros, Nydevik & Viitanen, 2003; Song et al., 2012). We applied a formula for sample size calculation for descriptive studies with proportions. Using a 30% prevalence rate, a precision value of 0.05 and a Z-value of 1.96, the formula yielded a minimum sample size of 323. Taking into account a 10% loss to follow up, this study planned to include a total of 355 participants.

Statistical analysis

Continuous variables and categorical variables were analyzed using independent t-tests, the Mann–Whitney test, Kruskal–Wallis test, chi-square test, or Fisher’s exact test, as appropriate. The Shapiro–Wilk test was performed for evaluating normality, and continuous variables in our dataset were normally distributed except for the NIHSS. Univariate and multivariate binary logistic regression and ordinal logistic regression analyses were performed to investigate the association of plasma OPG levels with poor functional outcomes. For multivariate analyses, sex, body mass index, and variables with p values < 0.1 in the univariate analysis (age, NIHSS, thrombolysis methods, number of trials for thrombectomy, successful recanalization, any hemorrhagic transformation, blood glucose level at admission, hemoglobin, total cholesterol, C-reactive protein, and vitamin D 25(OH)D) were adjusted. Because of over-fitting in the logistic regression, hemorrhagic transformation was entered after dichotomization. For the sensitivity analysis, plasma OPG levels were entered in the multivariate analysis model as continuous variables, per standard deviation and categorical variable (tertiles), respectively. For investigating the predictability of the plasma OPG levels for the prognosis, receiver operator characteristic (ROC) curves, area under the curve (AUC), integrated discrimination index (IDI) and net reclassification improvement (NRI) were performed to evaluate the performance of OPG levels for predicting poor outcomes (Pencina, D’Agostino Sr & Demler, 2012). We performed a subanalysis to check whether the association between OPG levels and functional outcome was consistent according to the door-to-puncture time. Using 8 h of door-to-puncture time as a cut-off, we divided patients into two groups (Jahan et al., 2019). All variables with p values < 0.05 were considered statistically significant. Statistical analyses were performed using SPSS Statistics for Windows, version 21.0 (IBM Corp., Armonk, NY, USA).

Results

Demographic data

Demographic data are shown in Table 1 and thrombectomy related variables and stroke subtypes are shown in Table 2. Of the 360 patients enrolled, 185 (51.4%) were men. Their mean age was 74.9 ± 13.9 years. Their mean ± standard deviation plasma OPG level was 200.2 ± 74.4 pg/mL. Of all included patients, 215 (59.8%) and 145 (40.2%) patients had good (mRS ≤2) and poor (mRS > 2) functional outcomes, respectively. Plasma OPG levels were higher in patients with poor outcomes (212.9 ± 63.7 vs. 191.7 ± 79.8 pg/mL, p = 0.008) than in patients with good outcomes. Patients with poor outcome were older (78.1 ± 13.2 vs. 72.7 ± 13.9 years, p < 0.001) and were more likely to have diabetes mellitus (68.3% vs. 41.9%, p < 0.001), undergo mechanical thrombectomy only (67.6% vs. 54.4%, p = 0.017), and undergo hemorrhagic transformation (46.9% vs. 28.8%, p = 0.001), and they less frequently underwent successful recanalization (74.5% vs. 94.4%, p < 0.001) than patients with good outcomes. Compared with patients with good outcomes, patients with poor outcomes also had higher NIHSS scores (median 18, interquartile range [13–22] vs. median 12 interquartile range [7–16], p < 0.001) and a higher mean blood glucose level at admission (148.2 ± 48.1 vs. 134.8 ± 50.9 years, p = 0.013). Patients with poor outcomes had lower mean vitamin D 25(OH)D (19.4 ± 6.7 vs. 21.5 ± 6.8 years, p = 0.004) and hemoglobin (12.9 ± 2.2 vs. 13.5 ± 1.8 years, p = 0.004) levels than patients with good outcomes (Tables 1 and 2). Table S1 shows the correlations between the plasma OPG concentration tertiles and other variables. Higher plasma OPG concentrations were associated with higher NIHSS scores (p = 0.011).

Table 1 Demographics of the included patients.

Variables	Total patients
(n = 360)	mRS ≤ 2
(n = 215)	mRS > 2
(n = 145)	p value	
Demographics					
Sex, male	185 (51.4)	113 (52.6)	72 (49.7)	0.665	
Age, years	74.9 ± 13.9	72.7 ± 13.9	78.1 ± 13.2	<0.001	
Body mass index, kg/m2	20.5 ± 4.0	20.8 ± 3.8	20.1 ± 4.2	0.102	
Risk factors					
Hypertension	269 (74.7)	163 (75.8)	106 (73.1)	0.648	
Diabetes mellitus	189 (52.5)	90 (41.9)	99 (68.3)	<0.001	
Hypercholesterolemia	164 (45.6)	95 (44.2)	69 (47.6)	0.598	
Coronary artery disease	97 (26.9)	57 (26.5)	40 (27.6)	0.917	
Congestive heart failure	27 (7.5)	12 (5.6)	15 (10.3)	0.139	
Atrial fibrillation	182 (50.6)	106 (49.3)	76 (52.4)	0.637	
Smoking	58 (16.1)	33 (15.3)	25 (17.2)	0.739	
Alcohol intake	87 (24.2)	49 (22.8)	38 (26.2)	0.537	
Previous stroke history	104 (28.9)	58 (27.0)	46 (31.7)	0.392	
Prior medication					
Anti-platelet	119 (33.1)	70 (32.6)	49 (33.8)	0.808	
Anti-coagulant	72 (20.0)	48 (22.3)	24 (16.6)	0.179	
Statins	117 (32.5)	69 (32.1)	48 (33.1)	0.841	
NIHSS	14 [9 –19]	12 [7 –16]	18 [13 –22]	<0.001	
Blood laboratory findings					
Osteoprotegerin, pg/mL	200.2 ± 74.4	191.7 ± 79.8	212.9 ± 63.7	0.008	
Vitamin D 25(OH)D, ng/mL	20.7 ± 6.9	21.5 ± 6.8	19.4 ± 6.7	0.004	
Glucose at admission, mg/dL	140.2 ± 50.1	134.8 ± 50.9	148.2 ± 48.1	0.013	
Triglyceride, mg/dL	112.0 ± 66.4	107.8 ± 67.0	118.3 ± 65.0	0.141	
Total cholesterol, mg/dL	163.5 ± 43.8	166.9 ± 42.8	158.5 ± 44.9	0.075	
Low-density lipoprotein, mg/dL	95.4 ± 36.4	96.9 ± 35.4	93.3 ± 37.9	0.353	
White blood cell count, ×103	8.3 ± 4.0	8.0 ± 3.9	8.7 ± 4.2	0.092	
Hemoglobin, mg/dL	13.3 ± 2.0	13.5 ± 1.8	12.9 ± 2.2	0.004	
Creatinine, mg/dL	0.9 ± 0.5	0.9 ± 0.5	0.9 ± 0.6	0.830	
Total calcium, mg/dL	8.2 ± 0.4	8.2 ± 0.4	8.2 ± 0.3	0.879	
Phosphate, mg/dL	3.1 ± 0.5	3.1 ± 0.5	3.1 ± 0.6	0.246	
C-reactive protein, mg/L	0.8 ± 2.2	0.6 ± 0.9	1.1 ± 3.3	0.026	
Notes.

Data are shown as n (%), mean ± standard deviation or median [interquartile range].

Statistical analyses were performed using the Chi-square test for categorical variables and independent t-test for continuous variables except NIHSS, analyzed by the Mann–Whitney test.

NIHSS, National Institute of Health Stroke Scale.

Table 2 Thrombectomy related variables and stroke subtypes of included patients.

Variables	Total patients
(n = 360)	mRS ≤2
(n = 215)	mRS>2
(n = 145)	p value	
Thrombolysis methods				0.017	
Mechanical thrombectomy only	215 (59.7)	117 (54.4)	98 (67.6)		
tPA and mechanical thrombectomy	145 (40.3)	98 (45.6)	47 (32.4)		
Onset-to-puncture time (min)	338.5 ± 300.2	332.5 ± 290.9	347.4 ± 314.2	0.645	
Number of trials for thrombectomy	2 [1–3]	1 [1–3]	2 [1–3]	0.058	
Recanalization (TICI IIb or III)	311 (86.4)	203 (94.4)	108 (74.5)	<0.001	
Hemorrhagic transformation				<0.001	
No hemorrhagic transformation	230 (63.9)	153 (71.2)	77 (53.1)		
HI1	45 (12.5)	29 (13.5)	16 (11.0)		
HI2	37 (10.3)	20 (9.3)	17 (11.7)		
PH1	22 (6.1)	7 (3.3)	15 (10.3)		
PH2	26 (7.2)	6 (2.8)	20 (13.8)		
Any hemorrhagic transformation	130 (36.1)	62 (28.8)	68 (46.9)	0.001	
Stroke subtype				0.564	
Cardioembolism	190 (52.8)	111 (51.6)	79 (54.5)		
Large artery atherosclerosis	58 (16.1)	34 (15.8)	24 (16.6)		
Undetermined two or more causes	48 (13.3)	34 (15.8)	14 (9.7)		
Undetermined negative	44 (12.2)	25 (11.6)	19 (13.1)		
Other determineda	20 (5.6)	11 (5.1)	9 (6.2)		
Notes.

Data are shown as n (%), mean ± standard deviation or median [interquartile range].

Statistical analyses were performed using the Chi-square test for categorical variables and independent t-test for continuous variables.

a Other determined stroke etiology is like dissection, endocarditis, or hypercoagulable state.

mRS modified Rankin Scale

tPA tissue plasminogen activator

TICI thrombolysis in cerebral infarction

HI hemorrhagic infarction

PH parenchymal hematoma

Plasma OPG levels and functional outcomes

After adjusting for sex, body mass index, and variables with p < 0.1 in the univariate analysis (age, diabetes mellitus, NIHSS, thrombolysis methods, number of trials for thrombectomy, recanalization, any hemorrhagic transformation, glucose at admission, hemoglobin, total cholesterol, and C-reactive protein, and vitamin D 25(OH)D), higher plasma OPG levels were independently associated with poor functional outcomes (continuous variable: odds ratios (OR) 1.004, 95% confidence interval (CI) (1.000–1.008), p = 0.031; per standard deviation: OR 1.373, 95% CI (1.039–1.824), p = 0.031; highest tertile vs. lowest tertile: OR 2.121, 95% CI [1.089–4.191], p = 0.037) in the multivariate binary logistic regression analysis (Table 3). Table S2 shows the associations between the poor functional outcome and other variables after adjusting plasma OPG concentration in the univariate and multivariate binary logistic regression analysis.

Table 3 Multivariable binary logistic regression analysis for the association of osteoprotegerin levels with poor functional outcomesa.

Variables	Adjusted OR (95% CI)	Log - odds	p value	
Osteoprotegerin (continuous variable)b	1.004 (1.000–1.008)	0.004	0.027*	
Osteoprotegerin per 1 SDb	1.373 (1.039–1.824)	0.317	0.027*	
Osteoprotegerin (categorical variable)b				
Tertile 1	Reference			
Tertile 2	1.990 (1.002–4.006)	0.688	0.051	
Tertile 3	2.121 (1.089–4.191)	0.752	0.028*	
Notes.

Data are shown as OR (95% CI).

* p < 0.05.

OR odds ratio

CI confidence interval

SD standard deviation

a Poor functional outcomes were defined by mRS ratings > 2.

b Adjusted for sex, body mass index, and variables with p values < 0.1 in the univariate analysis (age, NIHSS, DM, thrombolysis methods, number of trials for thrombectomy, successful recanalization, any hemorrhagic transformation, blood glucose level at admission, hemoglobin, total cholesterol, WBC, C-reactive protein, and vitamin D 25(OH)D).

In the ordinal logistic regression analysis with mRS as a dependent variable, higher plasma OPG concentrations were associated with poor functional outcomes (continuous variable: OR 1.004, 95% CI [1.002–1.007], p = 0.029; per standard deviation: OR 1.388, 95% CI [1.134–1.700], p = 0.029; highest tertile vs. lowest tertile: OR 2.102, 95% CI [1.301–3.412], p = 0.002) (Table 4). Table S3 shows the associations between the poor functional outcome and other variables after adjusting plasma OPG concentration in the univariate and multivariate binary logistic regression analysis. In the ROC comparison, the AUC of the multivariate analysis model with OPG was higher, but not significantly so, than the model without OPG (AUC 0.859 vs 0.855, p = 0.379). In contrast, NRI was significantly increased in the multivariate analysis model with OPG relative to without OPG (0.012, p = 0.027), although IDI did not increase it (0.174, p = 0.104) (Table S4).

Table 4 Multivariable ordinal logistic regression analysis for the association of osteoprotegerin levels with poor functional outcomesa.

Variables	Adjusted OR (95% CI)	Log - odds	p value	
Osteoprotegerin (continuous variable)b	1.004 (1.002–1.007)	0.004	0.002*	
Osteoprotegerin per 1 SDb	1.388 (1.134–1.700)	0.328	0.001*	
Osteoprotegerin (categorical variable)b				
Tertile 1	Reference			
Tertile 2	2.232 (1.358–3.690)	0.803	0.002*	
Tertile 3	2.102 (1.301–3.412)	0.743	0.003*	
Notes.

Data are shown as OR (95% CI).

* p < 0.05.

OR odds ratio

CI confidence interval

SD standard deviation

a Poor functional outcomes were defined by mRS ratings > 2.

b Adjusted for sex, body mass index, and variables with p values < 0.1 in the univariate analysis (age, NIHSS, DM, thrombolysis methods, number of trials for thrombectomy, successful recanalization, any hemorrhagic transformation, blood glucose level at admission, hemoglobin, total cholesterol, WBC, C-reactive protein, and vitamin D 25(OH)D).

In the subgroup analysis, there were no associations between plasma OPG levels and functional outcomes at 3 months after index stroke according to sex, age, hypertension, diabetes mellitus status, body mass index, hypercholesterolemia status, coronary artery disease status, congestive heart failure status, atrial fibrillation status, smoking history, alcohol intake, previous stroke history, NIHSS score, or thrombolysis method (mechanical thrombectomy only vs. tPA and mechanical thrombectomy). Onset-to-puncture time (<480 min vs. ≥480 min, p for interaction = 0.028) was the only variable with a significant association in the subgroup analysis. In the binary logistic regression analysis, the association of plasma OPG levels with poor functional outcomes was significant among patients with door-to-puncture times < 480 min (OR 1.551, 95% CI [1.199–2.030], p = 0.001), but it was not significant in patients with door-to-puncture times ≥480 min (OR 0.863, 95% CI [0.541–1.355], p = 0.526) (Table S5).

Discussion

The key finding of this study was that higher plasma OPG levels were associated with poor functional outcomes in acute ischemic stroke patients who underwent EVT. Several previous studies have revealed that OPG levels are associated with the presence and severity of carotid artery stenosis, coronary atherosclerosis, and peripheral artery atherosclerosis (Helske et al., 2007; Kadoglou et al., 2008; Rhee et al., 2005). Previous studies also showed that OPG might predict symptomatic carotid atherosclerosis (Musialek et al., 2013) and that OPG is a risk factor for progressive atherosclerosis and cardiovascular disease (Kiechl et al., 2004). The Framingham Heart Study confirmed that higher plasma OPG levels are associated with increased severity of silent lacunar infarction and white matter hyperintensities in brain MRI (OR 1.1, 95% CI [1.0–1.2]) (Shoamanesh et al., 2015). Moreover, stroke patients with severe cerebral artery atherosclerosis and poor functional outcomes have higher plasma OPG levels (Kim et al., 2013; Song et al., 2012; Ustundag et al., 2011). Indeed, there are significant associations between coronary artery disease, higher plasma OPG levels, and long-term mortality in the context of coronary artery disease (Omland et al., 2008; Venuraju et al., 2010) and future cardiovascular death in the general population (Vik et al., 2011). Long-term all-cause mortality in stroke patients is also associated with higher plasma OPG levels (Jensen et al., 2010). Our findings are in line with these previous studies and may provide additional evidence that plasma OPG levels are associated with poor functional outcomes among patients who have undergone EVT.

Our study could not delineate the mechanism underlying the association between increased plasma OPG levels and poor functional outcomes, but some hypotheses have been proposed. First, higher OPG levels are associated with modulations of matrix metalloproteinase-9 production in vascular cells, and matrix metalloproteinases (including matrix metalloproteinase-9) play an important role in neuroinflammation in patients undergoing thrombolytic therapy (Heo et al., 2003). Second, mechanical thrombectomy is associated with endothelial trauma or injury (Park et al., 2013). The OPG pathway also can modulate endothelial dysfunction and endothelial inflammation (Rochette et al., 2019; Shin, Shin & Chung, 2006). Therefore, the OPG-related molecular pathway could be activated by endothelial injury or endothelial dysfunction caused by brain damage or mechanical thrombectomy itself. Third, OPG can induce a complex with von Willebrand factor in endothelial vessels (Zannettino et al., 2005). This OPG–von Willebrand factor complex is presented in blood and could modulate vascular injury, inflammation, and thrombogenesis (Zannettino et al., 2005). Therefore, increased endothelial destabilization and thrombogenesis due to brain tissue injury or mechanical thrombectomy via OPG–von Willebrand factor complexes may contribute to poor functional outcomes. Fourth, it is known that calcified plaque lesions express RANKL, which initiates osteogenic phenotypic transformation and increases alkaline phosphatase activity. OPG, which functions as a decoy receptor for RANKL, may inhibit this osteogenic process and thus arterial calcification. Therefore, increased OPG levels may reflect the degree of calcified plaque lesions (Venuraju et al., 2010).

In this study, the association between plasma OPG levels and poor functional outcomes among patients who underwent EVT was demonstrated specifically among patients with door-to-puncture times < 480 min. We acknowledge that we cannot clearly explain why plasma OPG levels are associated with poor functional outcomes only among patients with an onset time < 8 h. Perhaps, for acute ischemic stroke patients who have undergone EVT, it can be speculated that the association between plasma OPG and functional outcomes varies over time. This may be because the effect of plasma OPG on stroke prognosis becomes smaller as time passes after stroke occurrence or because other factors other than OPG have greater effects on prognosis. Further studies are needed in this area. Therefore, when interpreting our study results, it should be noted that the relationship between plasma OPG levels and poor functional outcomes in acute ischemic stroke patients who underwent EVT was significant mainly in patients with door-to-puncture times < 480 min.

There were some limitations to this research. First, we did not have blood sample data for the general population. However, this study investigated the association between plasma OPG levels and functional outcomes in stroke patients who underwent EVT. Second, all examination results were collected once at the time of hospitalization. Therefore, we could not evaluate continuous or serial changes in plasma OPG levels or the effects of such changes on stroke prognosis. Third, it is difficult to infer causal relationships because our study is an observational study for the association.

This study showed that plasma OPG levels are related to prognosis in acute stroke patients receiving EVT. Accordingly, we could utilize plasma OPG as a prognostic biomarker in stroke patients undergoing EVT. For example, if the plasma OPG level is high in stroke patients after they undergo EVT, the outcome may be poor, so these patients may need a more aggressive treatment strategy or close monitoring. Furthermore, the possibility of endothelial damage can be estimated based on the OPG level, and it can be used as a reference for limiting the number of trials for stent retrievers in EVT.

Conclusions

Our study demonstrated that increased plasma OPG levels were associated with poor functional outcomes in acute ischemic stroke patients who underwent EVT. We may ascribe this association to the pleiotropic roles of OPG in neuroinflammation, endothelial dysfunction and thrombogenesis. Plasma OPG may be a potential biomarker for predicting neurologic outcomes in acute ischemic stroke patients who undergo EVT.

Supplemental Information

Supplemental Information 1 Correlation of tertile for the plasma OPG concentrations with other variables

Click here for additional data file.

Supplemental Information 2 Multivariate binary logistic regression analysis for the association of osteoprotegerin levels with functional outcomes

Click here for additional data file.

Supplemental Information 3 Multivariate ordinal logistic regression analysis for the association of osteoprotegerin levels with functional outcomes

Click here for additional data file.

Supplemental Information 4 Comparison of model fitness

Click here for additional data file.

Supplemental Information 5 Multivariable binary logistic analysis for association of osteoprotegerin levels with functional outcome

Click here for additional data file.

Supplemental Information 6 Supplementary Methods

Click here for additional data file.

Supplemental Information 7 Raw data

All risk factors analyzed for confirming relationship plasma osteoprotegerin levels and function outcome in stroke patients received endovascular thrombectomy.

Click here for additional data file.

Supplemental Information 8 Codebook

Click here for additional data file.

Additional Information and Declarations

Competing Interests

Author Contributions

Ethics

Data Availability

The authors declare there are no competing interests.

Moo-Seok Park conceived and designed the experiments, authored or reviewed drafts of the paper, and approved the final draft.

Jin-Hee Park performed the experiments, authored or reviewed drafts of the paper, and approved the final draft.

Ahran Joo performed the experiments, prepared figures and/or tables, and approved the final draft.

Yoonkyung Chang conceived and designed the experiments, analyzed the data, prepared figures and/or tables, and approved the final draft.

Tae-Jin Song conceived and designed the experiments, analyzed the data, prepared figures and/or tables, authored or reviewed drafts of the paper, and approved the final draft.

The following information was supplied relating to ethical approvals (i.e., approving body and any reference numbers):

This study was approved by our Institutional Review Board (EUMC 2014-04-023, 2018-10-036).

The following information was supplied regarding data availability:

The raw data is available in the Supplementary File.

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
