# Peer review of "The association of plasma osteoprotegerin levels and functional outcomes post endovascular thrombectomy in acute ischemic stroke patients: a retrospective observational study"

_PeerJ, doi:10.7717/peerj.13327_

## Round 0.1 · original submission · Major Revisions

1. Provide a more detailed but clear concise objective in the last sentence of the Introduction.

2. Reformat Tables 2 and 3. They are difficult to read in their present form

3. Review these articles https://www.ahajournals.org/doi/full/10.1161/01.CIR.0000127957.43874.BB and https://www.sciencedirect.com/science/article/pii/S0735109710010910 : add to your discussion

4. Write small n to replace capital N (fig 1)

5. Look at https://www.sciencedirect.com/science/article/pii/S0735109710010910 to guide writing your conclusion (but don't make it too long).

Reviewer 1 ·

Basic reporting

The paper is clearly written and well organized.

Experimental design

1. In general, this primary research is within the aims and scope of the journal. The experimental design was clearly written in a structured manner.
2. However, I suggest the authors describing the assessment of outcome variable, i.e., mRS score (Line 125), e.g., by whom (inter-rater reliability if there is more than one assessor), using what platform (face-to-face visit, or through teleconsultation). This description is closely related to the potential information bias in the study.
3. More information is needed to show that this research was conducted in conformity with the prevailing ethical standards in the field, e.g., the issue of informed consent during the data collection process, impact of withdrawal from participation, methods to ensure privacy and confidentiality in the stage of data management.

Validity of the findings

1. The data analysis is statistically sound, however insufficient information to report the specific selection of test (Table 1). I suggest the authors using footnote to represent which p-value was obtained from chi-square test, independent t-test, etc.
2. Also, the information about the normality of the continuous variables is contradicting. Line 142-143 (continuous variables in our dataset were normally distributed) contradicts with the presentation of NIHSS scores in median and interquartile range (Line 169 and Table 2).
3. Parameters showing model fitness e.g., area under ROC curve might be useful for model comparison in multivariable binary logistic regression analysis (Table 2).
4. It is interesting to include the original logistic regression analysis with mRS (the original mRS score without dichotomization) as a dependent variable, perhaps the authors could present that result (line 186).
5. The conclusions are not comprehensive, in which it could be linked to the research gap highlighted (Line 57-58) and how the study contributes to filing that gap e.g., in the current clinical practice of in-hospital stroke management with EVT.

Additional comments

This is a well-conducted research, and its contribution is desirable for future work.

Reviewer 2 ·

Basic reporting

Title of the study: Rewrite the title as it already stated the finding rather than the hypothesis of the study. It is suggested to modify it into “Association of Plasma Osteoprotegerin levels with Functional Outcomes in Acute Ischemic Stroke Patients who have undergone Endovascular Thrombectomy”

Abstract: Simplified the result and strengthen the methodology input.

Introduction: The information aims to justify the rationale of conducting the study. However, we notice that the info is inadequate. Readers want to know more about the association between OPG and stroke, especially during the acute phase. How these conditions can be associated with and determined the prognosis of the stroke patient? Does the level of plasma OPG change during EVT? What are the conditions that cofounded the level of OPG?

Experimental design

Method and tool: This section needs to be elaborated in detail and relevant to the topic. The authors need to show the robustness of the methodology conducted. Information related to study design, study location, study criteria, selection of the sample, and sample size determination is not mentioned in the manuscript. Please simplify the info under the subheading “Covariates” so that the readers could understand it better. Please elaborate on how the outcome of the study was measured. We notice that there is a discrepancy of information return in the text and in Figure 1. Please choose the correct one.

Validity of the findings

Result: Please explain in the legend what you mean by three different models in Table 2 and 3. How did you determine the fitness of those models?

Discussion: You do it well the discussion by citing the up-to-date references. However, we could not find any recommendation derive from your finding. Please provide significant recommendations that might help in managing acute stroke cases in the future. We also encourage you to give acknowledgment to those involved either directly or indirectly in the study.

Additional comments

Thank you for giving me the opportunity to review this interesting work. Your aim is to look at the association between plasma oesteoprotegrin level and functional outcome in stroke patients.
You did the writing very well but still, there are spaces for improvement.

Annotated reviews are not available for download in order to protect the identity of reviewers who chose to remain anonymous.

Reviewer 3 ·

Basic reporting

Adequate

Experimental design

Acceptable

Validity of the findings

Good findings

Additional comments

- Study design not mentioned in the title. Instead of writing the positive association in the title, suggest to revise the title to mention the objective that was mentioned in the last paragraph in the Introduction section
- Abstract is acceptable
- Introduction is adequate
- In Methods, please mention the type of study (Cross sectional)
- In Methods, to detail out steps taken to reduce sources of bias
- In Methods, to explain how study size was arrived at
- Results and Tables are adequate
- In Discussion, summarize the key findings in the first paragraph in reference to the study objectives
- To put in the Ethics clearance for this study and funding (if any)

---

## Round 0.2 · Minor Revisions

Abstract:
Write [From April 2014 through December 2020, a total of 360 acute ischemic stroke patients who underwent EVT were ...]

Main article:
Table 1:
Rephrase [aOther determined is like dissection, endocarditis, or hypercoagulable state.]. What does that mean?

Table 1 seems a bit long. If possible, could the authors split it into two tables? Make the first table for variables demographics, risk factors and prior medication. The following table is for the rest of the variables.

Table 2:
If Table 2 could be split into two tables and become Table 2 and Table 3, would it improve the presentation? For example, with two tables, the authors can add the log-odds values and the p-values.

Reviewer 1 ·

Basic reporting

No comment.

Experimental design

The methodology is very much improved, however there is room for improvement.
1. Objective - suggest to use the word 'associated/association' instead of 'positively correlated' (Line 82).
2. Study design - cross-sectional study isn't appropriate as the exposure (OPG) preceds the outcome (3-month mRS).
3. Subjects - information on sampling method is lacking (Line 95) and suggest to add the definition or diagosis of acute ischemic stroke; e.g., ICD -10 coding, or clinical diagnosis or radiological diagnosis (Line 108)
4. Subgroup analysis based on door-to-puncture time, suggest to justify the rational of conducting this subgroup analysis, preferably with supporting evidence - this should be mentioned under the analysis of subheading methods.
5. Results - suggest to report the standard deviation of age (Line 191), and you may consider baseline mrs as the covariate in the modelling.
6. 'positively associated' (Line 247) isn't appropriate, suggest to remove the word 'positively'

Validity of the findings

No comment.

Reviewer 2 ·

Basic reporting

Literature references, sufficient field background/context provided.

Experimental design

Methods described with sufficient detail & information to replicate. I am satisfied with all of the changes made.

Validity of the findings

All underlying data have been provided; they are robust, statistically sound, & controlled

Additional comments

The authors clarified all of the remarks, which I found to be satisfactory and appropriate.

Reviewer 3 ·

Basic reporting

Clear and corrections have been made

Experimental design

All issues commented in the previous review have been addressed

Validity of the findings

Valid and statistically sound

---

## Round 0.3 · Minor Revisions

- Write title as "The association of plasma osteoprotegerin levels and functional outcomes post endovascular thrombectomy in acute ischemic stroke patients: a retrospective observational study". Make sure all relevant changes made


- For Table 2, please improve it. For example, the first column (variables) is too crowded. The rows for columns 2, 3, and 4 are not aligned.

- For Table 3 and 4, in the 1st column, write the column name "Variables". Replace "Multivariate" with "Multivariable". In the 2nd column, replace "Binary logistic regression" with "Adjusted OR" and replace "Ordinal logistic regression" with "Adjusted OR".

---

## Round 0.4 · Minor Revisions

There is one small comment; Put additional footnotes in Table 3 and 4 to indicate the outcome variable and which level is the reference level (see attached PDF)

---

## Round 0.5 · accepted · Accept

I have decided to accept the manuscript; however, I hope the authors can make two small amendments to the caption and the footnote of Table 4 during proofreading (see attached file)